# Tumor-Induced T Cell Polarization by Schwann Cells

**DOI:** 10.3390/cells11223541

**Published:** 2022-11-09

**Authors:** Galina V. Shurin, Kavita Vats, Oleg Kruglov, Yuri L. Bunimovich, Michael R. Shurin

**Affiliations:** 1Department of Pathology, University of Pittsburgh Medical Center, Pittsburgh, PA 15261, USA; 2Department of Dermatology, University of Pittsburgh Medical Center, Pittsburgh, PA 15213, USA; 3Department of Immunology, University of Pittsburgh Medical Center, Pittsburgh, PA 15213, USA; 4Clinical Immunopathology UPMC, CLB, Room 4024, 3477 Euler Way, Pittsburgh, PA 15213, USA

**Keywords:** Schwann cells, melanoma, TGF-β, SMAD, ERK, T cells, T cell polarization

## Abstract

Nerve-cancer crosstalk resulting in either tumor neurogenesis or intratumoral neurodegeneration is critically controlled by Schwann cells, the principal glial cells of the peripheral nervous system. Though the direct stimulating effect of Schwann cells on malignant cell proliferation, motility, epithelial–mesenchymal transition, and the formation of metastases have been intensively investigated, the ability of Schwann cells to affect the effector and regulatory immune cells in the tumor environment is significantly less studied. Here, we demonstrated that tumor cells could stimulate Schwann cells to produce high levels of prostaglandin E, which could be blocked by COX-2 inhibitors. This effect was mediated by tumor-derived TGF-β as neutralization of this cytokine in the tumor-conditioned medium completely blocked the inducible prostaglandin E production by Schwann cells. Similar protective effects were also induced by the Schwann cell pretreatment with TGF-βR1/ALK4/5/7 and MAPK/ERK kinase inhibitors of the canonical and non-canonical TGF-β signaling pathways, respectively. Furthermore, prostaglandin E derived from tumor-activated Schwann cells blocked the proliferation of CD3/CD28-activated T cells and upregulated the expression of CD73 and PD-1 on both CD4+ and CD8+ T cells, suggesting T cell polarization to the exhausted phenotype. This new pathway of tumor-induced T cell inhibition via the activation of neuroglial cells represents new evidence of the importance of nerve–cancer crosstalk in controlling tumor development and progression. A better understanding of the tumor-neuro-immune axis supports the development of efficient targets for harnessing this axis and improving the efficacy of cancer therapy.

## 1. Introduction

The transforming growth factor-β (TGF-β) superfamily, in addition to the prototype TGF-β isoforms, includes more than 30 proteins, such as activins, nodal, growth differentiation and neurotrophic factors, and other cytokines and hormones. TGF-β transmits signals through two transmembrane serine-threonine kinase receptors, type I and type II (TβRI and TβRII) [1]. The TGF-β ligand first binds to TβRII, fosters phosphorylation of TβRI, and transmits intracellular signals in a SMAD-dependent (canonical) or -independent (noncanonical) manner [2]. Activation of non-SMAD signaling pathways, such as the MAP kinase pathway, in turn activates extracellular signal-regulated kinases (ERK1 and 2), p38, and JNK MAP kinases [3].

The TGF-β superfamily members play an imperative role in normal development and homeostasis as the key regulators of embryonal development, cellular differentiation, hormone secretion, wound healing, and tissue repair [4]. In the nervous system, TGF-β family members are necessary for proper neural development and function. During embryogenesis, they regulate the initial formation of the nervous system, synaptogenesis and plasticity, stem-cell lineage commitment to neuroglia and neurons, cell motility and survival, and axon guidance [5]. Later in life, TGF-β family members control neuroinflammatory responses, regulate quiescence in neural stem cells, and have a role in neural injury and neurodegenerative and neuropsychiatric diseases [6].

The TGF-βs are also powerful regulators of the immune system, pleiotropically affecting both cellular and antibody-mediated responses, as well as inflammatory reactions and immune tolerance [7]. In addition to controlling the expression of major histocompatibility (MHC) antigens and the activation, proliferation, effector functions, and apoptosis of immune and hematopoietic cells, TGF-β is also involved in pathogenesis of immune-mediated diseases including autoimmune and allergic diseases, infection and cancer [8,9,10,11]. TGF-βs proven to play an essential role in carcinogenesis and tumor progression. TGF-β, which is produced by both the malignant cells and tumor stromal and infiltrating cells, endorses epithelial–mesenchymal transition, expands cell invasiveness and metastatic progression, promotes angiogenesis, plays a central role in immune suppression and immune cell polarization within the tumor microenvironment, and participates in tumor immune evasion and poor responses to cancer immunotherapy [10,12]. However, TGF-β pathogenic roles and operational mechanisms in altering the tumor neuroenvironment are still generally unclear.

The presence and functional significance of the peripheral nervous system (PNS) in solid tumors in experimental animals and humans have been finally accepted supporting the usefulness of the term “the tumor neuroenvironment”. Accumulating data suggest that the existence and size of nerves within the solid tumor microenvironment are associated with tumor aggressiveness [13,14,15]. While the role of neurotransmitters and neuropeptides of the sympathetic and parasympathetic systems in supporting tumor growth and promoting tumor spreading is well documented [16,17,18], much less is known about the regulatory role of the PNS neuroglial cells, Schwann cells (SC), in the formation of the tumor neuroenvironment [19,20]. We have recently reported that tumor-activated SC directly up-regulate motility of lung cancer cells and increase formation of metastasis [21]. SC are also capable of attracting dendritic cells (DC) and myeloid-derived suppressor cells (MDSC) to the tumor site and augment their ability to inhibit T cell activation [20,22]. However, the effect of SC on T cells in cancer has not been investigated. How the tumor-modulated PNS neuroglial cells may affect T lymphocytes and whether tumor-derived factors can directly polarize glial cells to the immunosuppressive phenotype remain elusive.

Although the list of the effects of TGF-β on immunity and carcinogenesis is documented comprehensively and continues to expand, there still lacks a perfect understanding of its involvement in the tumor neuroenvironment and its role in regulating the neuro-immune axis in cancer. Here, we revealed that tumor-derived TGF-β and TGF-β-signaling are involved in the modulation of SC activity, which in turn results in inhibition of T cell activity and immunosuppression. These data uncover a novel protumorigenic pathway of PNS involvement in tumor progression and demonstrate how neuroglial cells, in addition to the PNS neurons, add to the formation and maintenance of the immunosuppressive tumor microenvironment.

## 2. Materials and Methods

### 2.1. Animals

C57BL/6 mice (7- to 8-week-old) from Taconic were housed in a pathogen-free facility under controlled temperature, humidity, and 12-h light/dark cycle with a commercial rodent diet and water available ad libitum. All studies were conducted in accordance with the NIH guidelines for the Care and Use of Laboratory Animals and approved by the Institutional Animal Care and Use Committee of the University of Pittsburgh.

### 2.2. Cell Lines and Cell Cultures

B16 murine melanoma cell line was purchased from ATCC. Mouse Ret melanoma cell line was a gift from V.Umansky (German Cancer Research Center, DKFZ, Heidelberg, Germany). Mouse BPWT melanoma cell line, derived from Braf^V600E^/Pten inducible melanoma model, was a gift from W. J. Storkus (University of Pittsburgh, Pittsburgh, USA). All primary and cell lines were authenticated, mycoplasma tested, were contaminant free, and used at low passage. Melanoma cell lines were cultured in a complete RPMI-1640 medium (GIBCO BRL – Thermofisher Scientific, Waltham, MA, USA) supplemented with 2 mM of L-glutamine, 100 U/mL of penicillin, 100 μg/mL of streptomycin, 10 mM HEPES (Invitrogen Life Technologies, Carlsbad, CA, USA), and 10% heat-inactivated fetal bovine serum (FBS, Gemini Bio-Products, USA). For the preparation of tumor-conditioned medium, subconfluent cultures were harvested, washed, and 1 × 10^6^ cells were cultured in 20 mL of complete RPMI 1640 medium with 10% FBS for 24 h. Then, the medium was replaced with medium supplemented with 2% FBS. After 24 h, conditioned medium was collected and centrifuged at 700× *g* at room temperature for 5 min to pellet the cells. The supernatant was collected and centrifuged at 2000× *g* at 4 °C for 10 min to remove cell debris.

Murine adult Schwann cells were prepared from the sciatic nerve of adult mice according to established protocols with some modifications as described earlier [20,21]. Briefly, the distal whole sciatic nerves were dissected under the stereoscopic microscope (Leica, Wetzlar, Germany) in sterile conditions. After 5-day incubation at 37 °C in SC medium (ScienCell Research Laboratories, Carlsbad, CA, USA), the sciatic nerves were dissociated using 1 mg/mL NB4G collagenase (MilliporeSigma, Burlington, MA, USA) in 3 mL HBSS (37 °C, 45 min). Then, the sciatic nerves were stripped free of epineurium, cut into 1-mm segments, and placed as explants into tissue culture dishes containing collagenase NB4G (37 °C, 30 min). After centrifugation (400× *g*, 5 min), the supernatant was discarded, the pellet was resuspended in 1 mL SC medium, and the nerves were triturated until they were completely dissociated. The cell suspension was diluted in SC medium supplemented with 2 μM forskolin (MilliporeSigma) and 50 ng/mL heregulin-β1 (MilliporeSigma) and cultured in Poly-d-lysine precoated flasks for 4 days following by an additional 6–10-day culture in the medium without forskolin and heregulin. During this time, SC were twice separated from fibroblasts using 300 μg/mL collagenase NB4G solution in 6 mL HBSS/75 cm^2^ flask (37 °C, 30 min). Prior to the experimentation, SC culture purity was established at 98–99% based on the immunofluorescent cytochemistry and flow cytometry phenotyping (BD LSR II) for SC-specific markers GFAP, p75^NGF^ and S100B. For the preparation of SC-conditioned medium, SC were treated with 10% (*v*/*v*) of tumor-conditioned medium or control medium for 24 h. Then, medium was removed and replaced with complete medium supplemented with 2% FBS for 24 h. Cell-free supernatant was collected by centrifugation.

### 2.3. Chemotaxis Assay

Chemotaxis assay for T cells migration was evaluated using a 48-well Transwell system (5 μm pore size, Corning Costar, Corning, NY, USA) with T cells placed in the upper chamber (1 × 10^6^ cells/mL, 100 μL). Assay medium (RPMI 1640 containing 2% FBS), CCL5/RANTES (20 ng/mL, PeproTech, Rocky Hill, NJ, USA) chemotactic for T cells (a positive control), conditioned medium from control SC and SC-treated with tumor-conditioned medium (B16, Ret and BPWT melanomas) were added to the bottom chamber (600 μL). Migration of T cells was assessed after 4-h incubation at 37 °C, 5% CO_2_. Cells transmigrated thought the membrane were collected and acquired on FACScan (Becton Dickinson, Franklin Lakes, NJ, USA) for 1 min for enumeration. Data are reported as the mean numbers of transmigrated cells from triplicate wells.

### 2.4. T Cell Analysis

T cells were isolated from single cell suspensions of mouse splenocytes by negative selection using Pan T cell isolation kit (Miltenyi Biotec, Bergisch Gladbach, Germany). Unwanted cells were targeted for removal with biotinylated antibodies directed against non-T cells and anti-biotin coated microbeads. Labeled cells were separated using magnetic column. Flow-through unlabeled cells were collected and contained the enriched T cells.

T cells proliferation was measured using a CFSE dye dilution assay. T cells (10 × 10^6^/mL) were labeled with 5µM CFSE (ThermoFisher Scientific, Waltham, MA, USA) followed by activation with beads coated with anti-CD3 and anti-CD28 for 24 h according to the manufacturer’s protocol (Life Technologies, Carlsbad, CA, USA). CFSE-labeled T cells (1 × 10^6^) were added to 24-well plates containing assay medium, control, and tumor-treated SC (2 × 10^4^) and incubated for 5 days. CFSE-labeled unstimulated T cells served as a control. T cells were harvested and analyzed by flow cytometry after labeling with antibodies to T cell specific markers: CD3-conjugated with APC (Biolegend, San Diego, CA, USA), CD4-conjugated with APC and CD8α-conjugated with PerCP/Cy5.5 (Biolegend). Proliferation of CFSE-labeled CD3+ cells was assessed by tracing dye dilution by flow cytometry. In additional experiments, expression of CD73, CD39 and PD-1 on control and SC-treated CD4 and CD8 T cells was measured using FITC-conjugated CD73 or CD39 antibody and PE-conjugated PD-1 antibody (Biolegend). Flow cytometric acquisition was performed on a Becton Dickinson LSR II instrument. Data were analyzed using FlowJo software (TreeStar Inc., San Francisco, CA, USA).

For assessing the role of PGE2 in inhibition of T cell proliferation and accumulation of regulatory T cells, SC (1 × 10^5^/mL in 12-well plates) were pre-treated for 2 h with 10 µM COX 2 inhibitor (FK-3311, Cayman Chemical Company, Ann Arbor, MI, USA) or vehicle control, then medium was removed and replaced with SC medium or tumor-conditioned medium (10% *v*/*v*) for 24 h and then replaced with medium supplemented with 2% FBS for 24 h. SC was seeded at 2 × 10^4^/0.5 mL in 24-well plates. After SC were adhered, 1 × 10^6^/0.5 mL T cells were added to SC culture and incubated for 5 days. T cell proliferation and phenotype were analyzed as describe above.

### 2.5. PGE2 Analysis

SC (1 × 10^5^/mL, 12-well plates) were pre-treated for 2 h with 10 µM COX-2 inhibitor (CAS 181696-33-3, Calbiochem, San Diego, CA, USA), 10 µM TGF-βR1/ALK4/5/7 inhibitor (SB431542, MilliporeSigma), 10 µM MAPK/ERK inhibitor (UO126, Cell Signaling Technology) or medium served as a control. Medium was removed and SC were treated with tumor-conditioned medium (10% *v*/*v*), control medium or *r*TGF-β1 (100 pg/mL, Cell Signaling Technology, Danvers, MA, USA) for 24 h. To block the TGF-β1 activity, 10 μg/mL anti-TGF-β1-neutralizing antibody (R&D System, Minneapolis, MN, USA) was added at the beginning of stimulation. Medium was removed and replaced with complete medium (2% FBS) for additional 24-h incubation. Cell-free supernatants were collected by centrifugation at 700× *g* for 5 min. PGE2 concentration (Cayman Chemical, USA) was measured in supernatants by ELISA.

### 2.6. TGF-β1 Production

Amounts of TGF-β1 secreted by B16, Ret, and BPWT melanoma cells and control and tumor-treated SC were measured in cell-free supernatants, collected as described above, using a commercial ELISA kit (ThermoFisher Scientific).

### 2.7. Western Blot Analysis

Cultured SC (1 × 10^6^) were treated with SC medium or B16-conditioned medium. Pellets were collected in 10, 20, 30, 60 and 180 min. Cells were lysed in RIPA buffer (Alfa Aesar, Ward Hill, MA, USA), normalized by protein concentration, separated on 4–12% SDS-PAGE, transferred to polyvinylidene difluoride membrane (Invitrogen) and analyzed with antibodies recognizing total SMAD1/2 and phospho-SMAD1/2 (1:1000, Cell Signaling Technology), total ERK1/2 and phosphor-ERK1/2 (1:1000, Cell Signaling Technology). For quantitation, gels were scanned and analyzed by UN-SCAN-IT gel software (Silk Scientific, Orem, UT, USA). The bands were quantified by pixel density using GAPDH as a housekeeping control. The results were analyzed as relative protein expression calculated as phosphorylated/total protein expression for each experimental group versus corresponding controls.

### 2.8. Quantitative PCR

Total RNA from control SC and SC treated with B16-conditioned medium was prepared using the Rneasy kit (Qiagen, Hilden, Germany). For reverse transcription, 1 µg of total RNA was primed with oligoDT (Roche Applied Science, Basel, Switzerland) for 10 min at 65 °C prior to the generation of cDNA via the addition of a master mix of reverse transcriptase 200 units MMLV-RT (Gibco/ThermoFisher Scientific), 1 mM dNTP (Promega, Madison, WI, USA) and 10 mM dithiothreitol per reaction for 1 h at 42 °C. The cDNA was then utilized for qPCR using standard qPCR procedures with the specific primers for COX2: Forward Primer AACCCAGGGGATCGAGTGT, Reverse Primer CGCAGCTCAGTGTTTGGGAT (IDT Integrated DNA Technologies, Coralville, IA, USA). GAPDH (Forward Primer AGGTCGGTGTGAACGGATTTG, Reverse Primer TGTAGACCATGTAGTTAGTTGAGGTCA) transcript was used as an internal control. Quantitative PCR on a Stratagene MX3000P procedure utilized SYBR green to monitor DNA synthesis. Reactions were run in triplicates in 25 µL reaction volume with 12.5 µL of SYBR master mix, 250 nM primers, and 1 µL of first strand cDNA. After 10 min at 95 °C, amplification involved 30 cycles of 30 sec at 95 °C, 30 sec at 56 °C and 1 min at 72 °C. The results are presented as relative transcription proportion to GAPDH.

### 2.9. Statistical Analysis

For a single comparison of two groups, the Student *t* test was used after the evaluation of normality. If data distribution was not normal, the Mann–Whitney rank-sum test was performed. For the comparison of multiple groups, ANOVA was applied. SigmaStat Software was used for data analysis (SyStat Software, Inc., Chicago, IL, USA). For all statistical analyses, *p* < 0.05 was considered significant. All experiments were repeated at least two times. Data are presented as the mean ± SEM.

## 3. Results

### 3.1. Polarization of T Cells by Tumor-Treated Schwann Cells

Purity and phenotypic characterization of cultured SC were validated as described earlier [21], and revealed high level of expression of neuroglia markers S100B, p75 and GFAP on tested cells (Appendix A). We also demonstrated that SC could chemoattract purified splenic T cells (Appendix A). Interestingly, melanoma B16-, ret- and BPWT-treated SC displayed a significantly higher ability to attract T cells compared to control SC (Appendix A). To confirm that control and tumor-treated SC express T cell chemokines, we determined chemokine expression and proved that melanoma cells were able to significantly up-regulate chemokine expression in SC, as shown in Appendix A. Based on these results, CCL5 was selected as a positive control for T cell attraction assay as demonstrated in Appendix A. Because tumor-treated SC were able to down-regulate inducible proliferation of T lymphocytes (Appendix A), we next asked whether SC could alter differentiation of T cells to regulatory/exhaustion phenotype.

We tested the impact of SC on T cell polarization by assessing expression of markers associated with T cell exhaustion, including PD-1, CD73, and CD39 [23,24]. We revealed that tumor-treated SC significantly up-regulated the expression of CD73 and PD-1 on CD8+ and CD4+ T cells up to 5–6 times (Figure 1 upper panels) but did not alter the expression of CD39 on T cells (Appendix A). For instance, while non-activated SC increased expression of CD73 on CD8+ T cells from 6.2 ± 0.9% to 9.4 ± 1.8%, tumor-activated SC up-regulated it to 26.5 ± 3.4% (*p* < 0.05). Expression of PD-1 on CD8+ T cells was increased from 8.6 ± 1.2% to 15.8 ± 2.2% by control SC and to 38.6 ± 6.3% by tumor-treated SC (*p* < 0.05). Similar up-regulation of CD73 and PD-1 was seen in CD4+ T lymphocytes. Remarkably, tumor-treated SC significantly suppressed the inducible proliferation of CD8+ and CD4+ T cells up to five folds (Figure 1, lower panels), suggesting their functional impairment, which may be associated with the exhausted phenotype. These results suggest tumor-activated SC can change both the function and phenotype of T lymphocytes, posing a question about SC factors responsible for this new phenomenon.

### 3.2. Schwann Cell-Derived Prostaglandin E2 Induces T Cell Polarization

As the expression of CD73 can be induced by PGE2 on macrophages [25] and PGE2 is a well-known regulator of T cell differentiation and plasticity [26], we first verified if adult SC could produce PGE2. First, analysis of mRNA expression of the inducible form of cyclooxygenase (COX-2), the major enzymes in prostaglandin production, revealed that SC express COX-2 mRNA and its level in SC could be significantly up to three-fold up-regulated by SC pre-exposure to the melanoma-conditioned medium (Figure 2A). PGE2 release in the SC-conditioned medium was directly determined by ELISA and these results demonstrated that tumor-treated SC produced significantly higher levels of PGE2 than control SC (Figure 2B). Importantly, PGE2 production could be blocked by a COX-2 inhibitor in both non-treated and tumor-treated SC, as shown in Figure 2C. As SC produce PGE2, we asked whether this prostaglandin may be responsible for the T cell exhaustion that can be induced by SC.

To answer this question, we treated SC with cell-permeable non-reversible non-toxic COX-2 inhibitor II-Benzenesulfonamide, 4-[5-(difluoromethyl)-3-phenyl-4-isoxazolyl] (CAS 181696-33-3), and compared its effect on SC with control-treated SC. As shown in Figure 3, blockage of PGE2 production by SC (see Figure 2C) completely abrogated upregulation of CD73 and PD-1 expression on CD8+ and CD4+ T cells induced by melanoma-treated SC to the control levels. In addition, COX-2 inhibitor blocked the ability of tumor-treated SC to suppress the inducible proliferation of T cells (Figure 3 low panels). For instance, while B16-activated SC inhibited proliferation of CD3CD28-activated CD8+ T cells from 39.3 ± 7.6% to 7.2 ± 3.2%, inhibition of PGE2 synthesis in SC returned T cell proliferation to the 34.7 ± 6.5% level (*p* < 0.05). Similar data were obtained for CD4+ T cells (Figure 3). Together, these results suggest that tumor-induced activation of SC to up-regulate production of PGE2 is responsible, at least partly, for the polarization of T cells to the exhausted phenotype. This raised the next question about potential tumor-derived factors which may activate the T cell-suppressive activity of SC.

### 3.3. Tumor-Derived TGF-β Activates Schwann Cells to Produce Prostaglandin E

The relationship between TGF-β and COX-2 has not been studied in SC. As some recent evidence suggests the involvement of TGF-β in nerve repair and regeneration, probably via reactivation of SC [27,28], and since evidence in other cell types had indicated the possible regulation of COX-2 by TGF-β [29,30], we put forward the hypothesis that TGF-β may upregulate PGE2 production in SC. Analysis of TGF-β expression by melanoma cell lines and cultured SC revealed that cells produced different levels of the cytokine and allowed selecting B16 cells as the highest source of TGF-β among tested cells (Figure 4A). Therefore, we next asked whether TGF-β may induce PGE2 production in SC in a way similar to B16 cells. Figure 4B demonstrates that both B16-conditioned medium and TGF-β activate SC to produce and release similar levels of PGE2. To verify that B16-derived TGF-β is responsible for PGE2 upregulation in SC, B16-conditioned medium was generated in the presence of neutralizing anti-TGF-β antibody. Figure 4C shows that neutralization of TGF-β significantly attenuated the ability of tumor cells to stimulate SC for PGE2 production. Together, these results demonstrate that tumor-derived TGF-β is able to activate SC to produce and release increased levels of PGE2. This raised the next question about the TGF-β-mediated signaling pathways in SC responsible for this new pathway of tumor-mediated SC activation.

### 3.4. Tumor-Induced Prostaglandin E Production by Schwann Cells Is Mediated by SMAD and ERK Signaling

TGF-β transduce intracellular signals by SMAD2 and SMAD3 proteins. TGF-β also induces non-SMAD signaling pathways, and at the intracellular signaling and transcription level, the ERK1/2 and SMAD pathways often associate with each other regulating the specific target genes in different cell types [3,31]. Analysis of these pathways in B16-activated SC revealed their role in mediating PGE2 production (Figure 5). First, Western blot analysis indicated that the level of the phosphorylation of SMAD 2/3 and ERK1/2 increased significantly in SC treated with tumor-conditioned medium (Figure 5A). Quantification of Western blots confirmed up-regulation of pSMAD2/3 and pERK1/2 up to 30 folds within 10 and 30 min, compared to tSMAD2/3 and tERK1/2, respectively (*p* < 0.001) (Appendix A). To prove that both canonical and non-canonical pathways in SC mediate PGE2 production induced by tumor treatment, we used SB431542, a selective inhibitor of the TGF-βRI/ALK4/5/7 canonical pathway, and UO126, a selective inhibitor of MAPK/ERK kinase, to treat SC prior to activation with B16-conditioned medium. MAPK/ERK kinase (MEK) is a major regulator of ERK1/2 signaling and can activate ERK1/2 transiently or persistently. Importantly, SB431542 has no effect on components of the ERK pathway. Results shown in Figure 5B indicate that both inhibitors significantly attenuated PGE2 production by SC induced by both B16 and TGF-β (*p* < 0.01). Together, our results suggest that tumor-derived TGF-β may activate SMAD and MAPK/ERK signaling pathways in SC, resulting in significant up-regulation of PGE2 production and release, inducing the exhaustion of T cells.

## 4. Discussion

SC, the main glial cells of the PNS, are now known as important cells in the formation and regulation of the tumor-neuro-immune axis that regulate tumor development and progression [19,20,32,33]. SC have been recently shown to play a role in maintenance of the immunosuppressive tumor microenvironment by chemoattraction and polarization of myeloid regulatory cells and upregulating their ability to reduce activity of T cells [20,22,34]. SC were also reported to be capable of directly stimulating malignant cell motility and formation of metastases [21,35,36]. However, SC-derived factors affecting myeloid regulatory cells have not been yet identified. Even less is known about tumor-derived agents that can alter activity and function of SC in the tumor milieu. Furthermore, a direct effect of SC on T cells in the tumor microenvironment has not been yet explored. Here, using several melanoma cell lines, we revealed that tumor-activated SC produce and release high levels of PGE2, which, in turn, induced exhausted phenotype in T cells and block their inducible proliferative activity. Many clinical and basic data emphasize PD-1 as a key co-inhibitory receptor in the development of T cell exhaustion [37,38]. Involvement of the PD-1 pathway has been shown to be a primary marker for exhausted T cells. For instance, PD-1 inhibits effector function and is broadly expressed in exhausted T cells [39]. Although CD39 and PD-1 surface markers may be not highly specific to exhaustion [40], the utility of detection of CD73 and PD-1 on regulatory/exhausted CD4+ and CD8+ T cells in combination with assessing T cell proliferative activity [41] allowed us to speculate that SC in the tumor milieu may affect T cell functionality.

CD73 ecto-5′-nucleotidase is expressed by cancer cells, endothelial cells and immune cells, predominantly T cells [42]. Regulatory T cells express high levels of CD73 and contribute to their inhibitory function in cancer by generating adenosine. However, the expression of CD73 and CD39 has also been demonstrated in non-regulatory T cells [43,44]. For instance, high CD73 expression in T cells has been reported to be associated with an exhausted phenotype of T cells [23]. Notably, CD73 on regulatory T cells impairs antitumor T cell responses [45]. Experimental studies with CD73-deficient mice revealed an immunosuppressive autocrine effect of CD73-mediated restraining of effector CD8+ T cell fitness and function [46]. Interestingly, new data suggest that CD8+CD73+ T cells may be especially important mediators of immunosuppression in human head and neck cancer [47]. Clinical data revealed a significant link between low frequencies of circulating CD73+CD8+ T lymphocytes and CD73+CD4+ regulatory T cells and better general survival of melanoma patients [48]. Although CD73 is essential to control inflammation under normal circumstances, it can support epithelial-to-mesenchymal transition, cell invasion, and angiogenesis in the context of the tumor microenvironment [49]. SC-induced T cell exhaustion seen as poor effector function and sustained expression of inhibitory receptors, may be an additional route of generation of a state of T cell dysfunction observed in chronic infection, chronic inflammation, and cancer [50,51]. Tumor-associated immunosuppression, including T cell exhaustion, continues to be a major barrier for effective cancer immunotherapy. Therefore, identification of all cellular components of the tumor environment responsible for blocking the antitumor potential of effector T cells is crucial for improving the efficacy of cancer treatment. Our data support the recent notion that the tumor neuroenvironment should be further investigated and tested as a new target for harnessing the tumor environ [20].

We have also revealed that tumor-derived TGF-β is an important cytokine that induces SC activation for PGE2 production via the activation of canonical and non-canonical signaling in SC. The expression of TGF-βRI and TGF-βRII on SC has been demonstrated earlier, suggesting that SC are therefore responsive to TGF-β1 and TGF-β3 [52]. In fact, the induction of SC activation and proliferation by TGF-β has been well documented, suggesting that TGF-β-induced SC dedifferentiation may enhance the regenerative response [53]. TGF-β1 may be important for the development of SC as it is expressed by immature SC and might affect SC longevity in vitro [54]. SC may also use TGF-βs to communicate with motoneurons [54]. However, the role of TGF-β in regulating SC-immune cell interactions has not been revealed in spite of the fact that TGF-β plays a crucial role in supporting the immunosuppressive tumor microenvironment. Our new data thus revealed a new target for TGF-β in cancer, Schwann cells, which can directly support malignant cell motility and affect immune regulatory and immune effector cells in cancer. Nevertheless, our results are limited by utilizing the melanoma cell lines and melanoma-derived TGF-β. Further studies are needed to evaluate this pathway for different types of cancer and to identify additional tumor-derived factors that might affect SC activity and their ability to regulate immune cells in the tumor microenvironment.

Prostaglandins may contribute to tumor development through several mechanisms including the regulation of malignant cell proliferation, apoptosis, metabolism or modulation of the immune responses [55]. For instance, COX-2 derived PGE2 can promote tumor growth by activating signaling pathways that control cell proliferation, motility, apoptosis and angiogenesis [56]. Interestingly, prostaglandins were found in the central nervous tissue where their various actions include effects on behavior, alteration of food intake, body temperature, and cardiovascular and motor functions [57,58,59]. Furthermore, PGE2 may have an important function in modulating the pro-inflammatory events happening in both vascular- and microglial-associated components at the injured site of the CNS [60]. In the PNS, prostaglandins may promote neurite outgrowth, participate in sensory perception, including pain sensation during inflammation, and influence the release of neurotransmitters from nerve endings [61,62,63,64]. However, the involvement of prostaglandins in regulating the tumor neuroenvironment and activity of the neuroimmune axis in the tumor milieu has not been investigated. Further studies may also determine how tumor-induced prostaglandin production by SC directly affects tumor cell growth and metastasis.

Some findings indicate that the reciprocal interaction of glial cell stimulation, COX enzymes and prostaglandins intermediates neurodegeneration and neuroprotection during brain inflammation [65]. PGE2 production has been shown in the malignant Schwann cell lines [66] and COX-2 expression was detected in SC after nerve injury [67]. Our data here demonstrate that although control SC generate low levels of PGE2, its production is significantly upregulated after treatment with tumor-conditioned medium or TGF-β and these effects can be blocked by COX-2 inhibitors. More importantly, SC-derived PGE2 can affect CD4+ and CD8+ T cells polarizing them to the exhaustion state. Many reports suggest that PGE2 changes polarization of T helper cells to different subtypes, alters T cell differentiation and plasticity and enhances the induction and differentiation of FoxP3 regulatory T cells [68]. Other studies indicate that PGE2 suppresses T cell proliferation, accelerates T cell replicative senescence markers, and induces T cell anergy [69], which could be a sign of exhaustion or induction of anergy mediated by PGE2. In concordance with such observations, our results allow suggesting a new mechanism of immunosuppression as a result of tumor–PNS interactions. These interesting observations may also add to the better understanding of the effects of COX-2 inhibitors on the chronic inflammatory tumor milieu as well as cancer prevention and treatment [70]. In addition, it will be important to determine whether tumor-derived TGF-β may alter the production of other immunomodulatory molecules by SC, such as cytokines and chemokines. Finally, further studies should also reveal if different types of malignant cells may control COX-2 activity in SC via common or cancer-specific factors.

In summary, we revealed that tumor-derived TGF-β and resulting canonical and non-canonical TGF-β signaling in SC caused their activation and prostaglandin production. In turn, SC-derived PGE2 suppressed T cell function and induced their exhausted phenotype, as shown by the upregulated expression of CD73 and PD-1. This new pathway of tumor-induced T cell inhibition via the activation of glial cells represents new evidence of the importance of nerve–cancer crosstalk in controlling tumor development and progression. A better understanding of how the tumor–neuro–immune axis operates should provide a foundation for developing efficient targets to harness this axis and improve the efficacy of cancer therapy.

## Figures and Tables

**Figure 1 cells-11-03541-f001:**
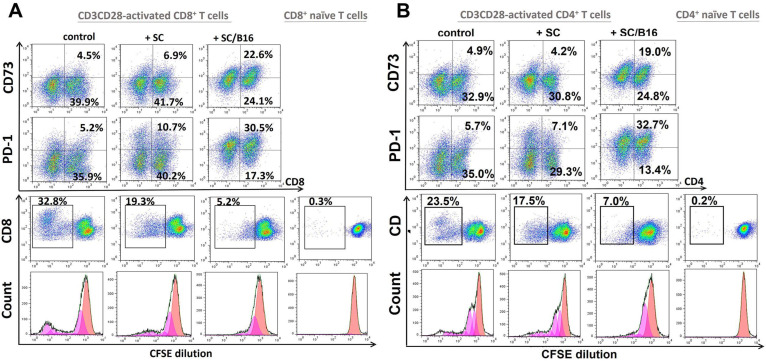
Tumor-activated Schwann cells induced exhaustion of T cells. Adult SC were isolated from the sciatic nerve of C57BL/6 mice, cultured and purified as described in M&M. SC were then co-cultured with medium (SC) or with B16-conditioned medium (10% *v*/*v*) for 48 h and washed. Then, control and melanoma-pretreated SC were mixed with SFSE-labeled splenic T cells that were activated by CD3-CD28-coated beads for 24 h. After 5 days, expression of CD73 and PD-1 on CD8+ (**A**) and CD4+ (**B**) T cells was evaluated by flow cytometry. Gating strategy was based on the initial selection of CD3+ T cells with the further analysis of CD73 and PD-1 expression on either CD8+ or CD4+ cells. Proliferation of CFSE-labeled T cells was assessed by tracing dye dilution by flow cytometry. Results from a representative experiment are shown.

**Figure 2 cells-11-03541-f002:**
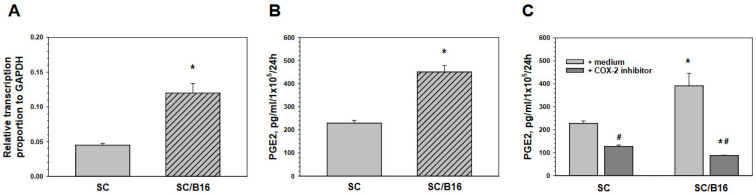
Malignant cells activated Schwann cells for PGE2 production. Adult SC were isolated from the sciatic nerve of C57BL/6 mice, cultured and purified as described in M&M. SC were then co-cultured with medium (SC) or with B16-conditioned medium (10% *v*/*v*) (SC/B16) for 24 h and washed. Control and tumor-treated SC were then assessed for COX-2 mRNA expression by qPCR (**A**) or cultured for another 24 h for measuring PGE2 production by ELISA (**B**). COX-2 inhibitor CAS181696-33-3 (10 µM, 24 h) prevented tumor-induced up-regulation of PGE2 expression in SC (**C**). Results are the mean ± SEM. *, *p* < 0.001 versus control SC group; #, *p* < 0.001 versus medium group (n = 3, Student *t*-test).

**Figure 3 cells-11-03541-f003:**
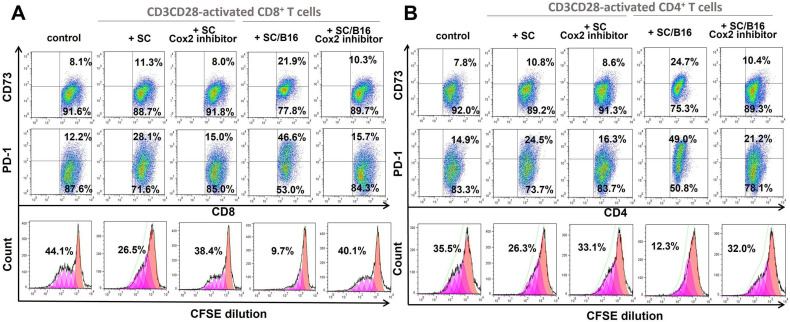
COX-2 inhibition prevented T cell exhaustion induced by tumor-activated SC. Adult SC were isolated from the sciatic nerve of C57BL/6 mice, cultured and purified as described in M&M. SC were then co-cultured with medium (SC) and B16-conditioned medium (10% *v*/*v*) (SC/B16) for 24 h after pretreatment with COX-2 inhibitor CAS 181696-33-3 (10 µM, 2 h). After washing, control and melanoma-pretreated SC were mixed with SFSE-labeled splenic T cells that were activated by CD3-CD28-coated beads for 24 h. After 5 days, expression of CD73 and PD-1 on CD8+ (**A**) and CD4+ (**B**) T cells was evaluated by flow cytometry. Gating strategy was based on the initial separation of CD4+ versus CD8+ cells with the consequent and independent analysis of each cell population for CD73 and PD-1 expression. Proliferation of CFSE-labeled T cells was assessed by tracing dye dilution by flow cytometry. Results from a representative experiment are shown.

**Figure 4 cells-11-03541-f004:**
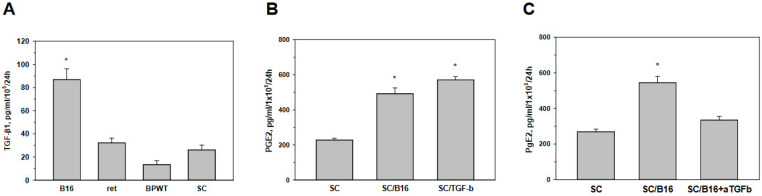
Tumor-derived TGF-β induced PGE2 production by Schwann cells. Adult SC were isolated from the sciatic nerve of C57BL/6 mice, cultured and purified as described in M&M. TGF-β production by melanoma B16, ret and BPWT cells and SC were assessed by ELISA in cell-free supernatant obtained from tumor cells and SC after they were washed and re-cultured for 24 h (**A**). SC were also co-cultured with medium (SC), B16-conditioned medium (10% *v*/*v*) (SC/B16) and 10 ng/mL TGF-β1 (SC/TGF-b) for 24 h, washed and cultured for additional 24 h. PGE2 levels were then determined in cell-free supernatants by ELISA (**B**). In addition, SC were cultured with medium (SC), B16-conditioned medium (10% *v*/*v*) (SC/B16) and conditioned medium from B16 cells that were cultured in the presence of anti-TGF-β neutralizing antibody (SC/B1b+aTFGb). Control and tumor-treated SC were then washed and cultured for another 24 h to assess PGE2 production by ELISA (**C**). Results are the mean ± SEM. *, *p* < 0.001 versus control SC group (n = 3, One-way ANOVA).

**Figure 5 cells-11-03541-f005:**
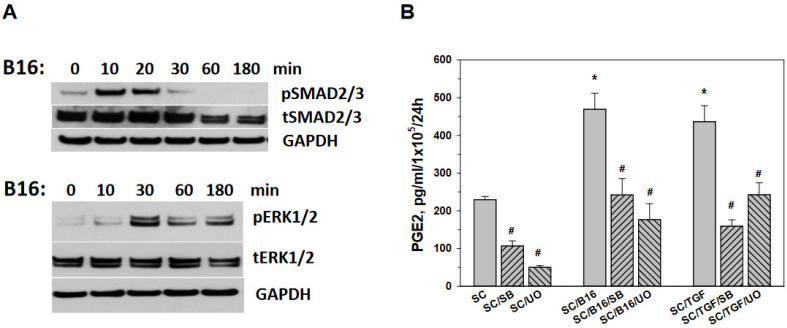
Tumor activated SC for PGE2 production via SMAD2/3 and ERK1/2 pathways. Adult SC were isolated from the sciatic nerve of C57BL/6 mice, cultured and purified as described in M&M. SC were then cultured with B16-conditioned medium (10% *v*/*v*) for 0–180 min, harvested, washed in cold medium and used for cell lysate preparation for protein expression analyses. Expression of phosphorylated (p) and total (t) SMAD2/3 and ERK1/2 proteins in control and tumor-treated SC was determined by Western blot analysis after proteins were separated by SDS-PAGE electrophoresis and immunoblotted as described in M&M. Representative Western blot results are shown (**A**). SC were also co-cultured with medium (SC), B16-conditioned medium (10% *v*/*v*) (SC/B16) and 10 ng/mL TGF-β1 (SC/TGF) for 24 h after pre-treatment with either 10 µM SB431542, a selective inhibitor of the TGF-βRI/ALK4/5/7 (SB) or 10 µM UO126, a selective inhibitor of MAPK/ERK kinase (UO). Pre-treatment with medium served as a control. Cells were then washed and cultured for additional 24 h. PGE2 levels were then determined in cell-free supernatants by ELISA (**B**). Results are the mean ± SEM. *, *p* < 0.001 versus control SC group; #, *p* < 0.001 versus B16- and TGF-β-treated SC groups, respectively (n = 3, One-way ANOVA).

## Data Availability

The data are contained within the article and Appendix A.

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
