# Peer review of "Tumor-Induced T Cell Polarization by Schwann Cells"

_cells, 2022, doi:10.3390/cells11223541_

Round 1

Reviewer 1 Report

This manuscript by Sharon et.al highlights the importance of nerve-cancer crosstalk, authors in this manuscripts provide compelling evidence of tumor cell regulation of Schwann cells via TGF-b to produce PGE2 and that leads to the polarization of T cells to the exhausasted phenotype.

Major comments-

Though this study clearly provide evidences that conditioned media from tumor cells can modulate nerve cells and exert subsequent effects on T Cells, however this is reductionist in vitro model approach and does not truly reflect tumor micro environment.    

Minor Comments-

1) Figure 2 A labelling of X-axis is not clear and same with figure legends of this figure, Line-258

2) Did the authors use the CD3 T cells in figure 1 and 3 as mentioned in figure legends? if yes then why there is difference in the flow plots of figure 1 and 3. In figure -1, there are two separate population for CD4 and CD8 while   Figure-3 has only positive population either of CD4 or CD8.

3)There are no quantification of western blots data in figure-5, though authors claim 1000 fold up regulation of p-SMAD2/3 and ERK1/2 which seems to be very far-fetched and does not reflect in respective western blots panel. line 319

4) Figure legends for Figure-5 are incomplete.

Author Response

Point-by-point answers to the reviewers’ concerns:

We want to thank all Reviewers for their helpful comments and suggestions!

Reviewer 1.

“Though this study clearly provide evidences that conditioned media from tumor cells can modulate nerve cells and exert subsequent effects on T Cells, however this is reductionist in vitro model approach and does not truly reflect tumor micro environment.”

We agree and want to mention that this study is a part of our research program focusing on the molecular mechanisms of cellular interactions in the tumor neuroenvironment. The goal of the manuscript is to provide an initial description of one of several pathways that involved in the formation of the tumor-neuro-immune axis. In doing so, we think it is important to reveal mechanisms of how tumor cell-derived TGF-β modulates the ability of Schwann cells to polarize T cell phenotype in vitro. Our establish in vitro model will allow verification of similar, although more complex, in vivo pathways we hope to identify in the following studies. And these experiments are in progress in our laboratory.  

“Figure 2 A labelling of X-axis is not clear and same with figure legends of this figure, Line-258”

We are sorry for the mistake in the Figure 2 legend text that occurred during the figure loading procedure. This mistake was corrected, and the revised version of Figure 2 legend is shown below:

Figure 2. Malignant cells activated Schwann cells for PGE2 production. Adult SC were isolated from the sciatic nerve of C57BL/6 mice, cultured and purified as described in M&M. SC were then co-cultured with medium (SC) or with B16-conditioned medium (10% v/v) (SC/B16) for 24 h and washed. Control and tumor-treated SC were then assessed for COX-2 mRNA expression by qPCR (A) or cultured for another 24 h for measuring PGE2 production by ELISA (B). COX-2 inhibitor CAS181696-33-3 (10 µM, 24 h) prevented tumor-induced up-regulation of PGE2 expression in SC (C). Results are the mean ± SEM. *, p<0.001 versus control SC group; #, p<0.001 versus medium group (n=3, Student t-test).

“Did the authors use the CD3 T cells in figure 1 and 3 as mentioned in figure legends? if yes then why there is difference in the flow plots of figure 1 and 3. In figure -1, there are two separate population for CD4 and CD8 while   Figure-3 has only positive population either of CD4 or CD8.”

We thank Reviewer 1 for mentioning this and are sorry for unclear explanation of our gating strategies for Figures 1 and 3. Our gating strategy on Figure 1 was based on the initial selection of CD3+ T cells with the further analysis of CD73 and PD-1 expression on either CD8+ or CD4+ cells. Our gating strategy in Figure 3 was based on the initial separation of CD4+ versus CD8+ cells with the consequent and independent analysis of each cell population (CD4+ and CD8+) for CD73 and PD-1 expression. This is why we see both positive and negative cells in Figure 1 (CD4+/CD4- and CD8+/CD8-) and only positive cells in Figure 3 (CD4+ or CD8+).

            We have added the explanation of our gating strategy to the figure legends in the revised version of the manuscript.

“There are no quantification of western blots data in figure-5, though authors claim 1000 fold up regulation of p-SMAD2/3 and ERK1/2 which seems to be very far-fetched and does not reflect in respective western blots panel. line 319”

We are sorry for omitting these results in the original version of the manuscript. For quantitation, gels were scanned and analyzed by UN-SCAN-IT gel software (Silk Scientific). The bands were quantified by pixel density using GAPDH as a housekeeping control. The results were analyzed as relative protein expression calculated as phosphorylated/total protein expression for each experimental group. Our original quantitative results were added as Figure S5. We also corrected data explanation and now show the relative expression of phospho-to-total SMAD2/3 and ERK1/2 expression instead of absolute increase of phosphorylated protein. Modified sentence in Results is shown below.

Quantification of Western blots confirmed up-regulation of pSMAD2/3 and pERK1/2 expression up to 30 folds within 10 and 30 min, compared to tSMAD2/3 and tERK1/2 expression, respectively (p<0.001) (Figure S6).

“Figure legends for Figure-5 are incomplete.”

We are sorry for another text-loading problem. Figure 5 legend was reloaded. Here is a corrected version of this figure legend:

Figure 5. Tumor activated SC for PGE2 production via SMAD2/3 and ERK1/2 pathways. Adult SC were isolated from the sciatic nerve of C57BL/6 mice, cultured and purified as described in M&M. SC were then cultured with B16-conditioned medium (10% v/v) for 0-180 min, harvested, washed in cold medium and used for cell lysate preparation for protein expression analyses. Expression of phosphorylated (p) and total (t) SMAD2/3 and ERK1/2 proteins in control and tumor-treated SC was determined by Western blot analysis after proteins were separated by SDS-PAGE electrophoresis as described in M&M. Representative Western blot results are shown (A). SC were also co-cultured with medium (SC), B16-conditioned medium (10% v/v) (SC/B16) and 10 ng/ml TGF-β1 (SC/TGF) for 24 h after pre-treatment with either 10 µM SB431542, a selective inhibitor of the TGF-βRI/ALK4/5/7 (SB) or 10 µM UO126, a selective inhibitor of MAPK/ERK kinase (UO). Pre-treatment with medium served as a control. Cells were then washed and cultured for additional 24 h. PGE2 levels were then determined in cell-free supernatants by ELISA (B). Results are the mean ± SEM. *, p<0.001 versus control SC group; #, p<0.001 versus B16- and TGF-β-treated SC groups, respectively (n=3, One-way ANOVA).

Reviewer 2 Report

In this study, Shurin et al. investigated the roles of tumor-Schwann cell cross-talk on T cell polarization using in vitro co-culture system. The authors discovered that tumor-treated Schwann cells can elevate the exhaustion markers of T cells via PGE2 secretion. Furtherly, they found tumor-driven TGF-Beta and SMAD2/3 and ERK1/2 pathways in Schwann cells play essential roles in Schwann cells. In short, the authors found a tumor-Schwann cell cross-talk axis is involved in T cell exhaustion, which is an interesting direction for tumor treatment. However, the authors should address the following points to make the whole work more convincing.

1. As the title is “Tumor-induced T cell polarization by Schwann cells”, the authors may try to address the general regulation for tumors to Schwann cells, however, the authors used 3 melanoma cell lines. Is this regulation melanoma-specific or generally existed in different tumors?

2. In Figure 2&3, the COX2 inhibitor can block the production of PGE2 and prevent T cell polarization induced by tumor-driven Schwann cells, is there any possibility that the COX2 inhibitor can affect Schwann cell vulnerability? 

3. In Figure 4, the melanoma ret and BPWT cells have very low secreted TGF-Beta1, do these 2 cell lines have effects on T cell polarization? 

4. In Figure 5A, why the phosphorylation levels of pSMAD2/3 and p-ERK1/2 decrease after  30 minutes of incubation with the B16-condition medium?

5. Please add the figure legend of Figure 5B.

Author Response

Point-by-point answers to the reviewers’ concerns:

We want to thank all Reviewers for their helpful comments and suggestions!

Reviewer 2.

“As the title is “Tumor-induced T cell polarization by Schwann cells”,the authors may try to address the general regulation for tumors to Schwann cells, however, the authors used 3 melanoma cell lines. Is this regulation melanoma-specific or generally existed in different tumors?”

We agree with the Reviewer that the title might look too broad. However, this is the generally accepted problem related to the common or tumor-specific mechanisms of tumor-induced modulation of different cell subsets in the tumor microenvironment. Although in this particular study we have used only murine melanoma cell lines, we plan to expand our future analyses to different cell lines. Because many different tumor cells produce TGF-beta, we may speculate that similar effects on Schwann cell can be seen in different model systems. At the same time, we have reported earlier that mouse lung cancer cell lines affect Schwann cells via different cytokines. Our newer data also revealed the role of microRNA in Schwann cell modulation by human lung cancer cells. In any way, we agree with the Reviewer that this is an important point, and we will try to focus on it in our future review publication.

“In Figure 2&3, the COX2 inhibitor can block the production of PGE2 and prevent T cell polarization induced by tumor-driven Schwann cells, is there any possibility that the COX2 inhibitor can affect Schwann cell vulnerability?”

This is an interesting and important point. To address it, we have used the treatment of Schwann cells with the inhibitor as a control. We did not see any significant alterations as shown in Figure 1. However, we will expand this potential pathway on other functions of Schwann cells since Schwann cell produce background levels of PGE2, and there are no data about possible paracrine and autocrine effects of PGE2 on these cells. We thank the Reviewer for this interesting idea.

“In Figure 4, the melanoma ret and BPWT cells have very low secreted TGF-Beta1, do these 2 cell lines have effects on T cell polarization?” 

This is also an important question, and we have addressed in in our initial studies. Yes, we saw an effect of ret and BPWT cells on Schwann cell-mediated T cell polarization, but it was significantly smaller. We know that TGF-beta is not the only tumor-derived factor that can induce activation of Schwann cells and, probably, production of PGE2. With our collaborators, we work on different mechanisms of Schwann cells activation and dedifferentiation in the tumor microenvironment and hope to reveal more tumor-specific and common factors that may affect Schwann cells and modulate their effect on different immune cells.

“In Figure 5A, why the phosphorylation levels of pSMAD2/3 and p-ERK1/2 decrease after 30 minutes of incubation with the B16-condition medium?”

This is a great question. We think that there are two main pathways that can explain this very common phenomenon, which was reported in different cellular systems. First, inactivation of active phosphorylated proteins by phosphatases. For instance, the activity of ERK is regulated by kinases and phosphatases. The regulatory dephosphorylation of ERK1/2, which plays a key role in regulating the magnitude and duration of kinase activation, is mediated by protein-tyrosine specific phosphatases, protein-serine/threonine phosphatases, and dual specificity phosphatases. The finding that multiple phosphatases inactivate ERKs suggests that the duration and extent of ERK activation is controlled by the balanced activities of kinases and phosphatases and represents an important feedback mechanism of ERK functioning (Theodosiou, A., Ashworth, A. MAP kinase phosphatases. Genome Biol 3, 2002; reviews3009.1)

The second known mechanism is the internalization of receptors: Receptor endocytosis is a pivotal regulatory mechanism in signal transduction. For instance, TGF-β receptors are internalized via both clathrin- and caveolae-dependent pathways and internalization of the TGF-β receptors via the clathrin-coated pits has been linked with signaling via SMAD2/3 and receptor recycling. In contrast, TGF-β receptor localization in caveolae is associated with downregulation of SMAD2/3 signaling and receptor degradation following ubiquitination by the E3-ubiquitin ligase Smurf2 (Albane et al. The TGF-β co-receptor, CD109, promotes internalization and degradation of TGF-β receptors. Biochimica et Biophysica Acta (BBA) - Molecular Cell Research. 2011; 1813 (5):742-753.)

“Please add the figure legend of Figure 5B.”

We are sorry for another text-loading problem. Figure 5 legend was reloaded. Here is a corrected version of this figure legend:

Figure 5. Tumor activated SC for PGE2 production via SMAD2/3 and ERK1/2 pathways. Adult SC were isolated from the sciatic nerve of C57BL/6 mice, cultured and purified as described in M&M. SC were then cultured with B16-conditioned medium (10% v/v) for 0-180 min, harvested, washed in cold medium and used for cell lysate preparation for protein expression analyses. Expression of phosphorylated (p) and total (t) SMAD2/3 and ERK1/2 proteins in control and tumor-treated SC was determined by Western blot analysis after proteins were separated by SDS-PAGE electrophoresis as described in M&M. Representative Western blot results are shown (A). SC were also co-cultured with medium (SC), B16-conditioned medium (10% v/v) (SC/B16) and 10 ng/ml TGF-β1 (SC/TGF) for 24 h after pre-treatment with either 10 µM SB431542, a selective inhibitor of the TGF-βRI/ALK4/5/7 (SB) or 10 µM UO126, a selective inhibitor of MAPK/ERK kinase (UO). Pre-treatment with medium served as a control. Cells were then washed and cultured for additional 24 h. PGE2 levels were then determined in cell-free supernatants by ELISA (B). Results are the mean ± SEM. *, p<0.001 versus control SC group; #, p<0.001 versus B16- and TGF-β-treated SC groups, respectively (n=3, One-way ANOVA).

Reviewer 3 Report

Introduction

The authors should briefly discuss the importance/role of prostaglandin in T-cell polarization.

Please make sure the text is readable and of good quality in all figures. Also, enhance the quality of figures.

Supplementary data

1)      Authors should put the ligands next to represent the figure, not on a separate page.

2)      It is better to provide the uncropped original image of gels/blots with the marker (authors may highlight the cropped portion(s) by line/box etc.).

3)      Original image of Figure A (pERK1/2 and tERK1/2) is missing.

Author Response

Point-by-point answers to the reviewers’ concerns:

We want to thank all Reviewers for their helpful comments and suggestions!

Reviewer 3

“The authors should briefly discuss the importance/role of prostaglandin in T-cell polarization.”

We agree and have expanded our discussion of this phenomenon. For instance, the following paragraph has been modified in the revised version of the manuscript:

… More importantly, SC-derived PGE2 can affect CD4+ and CD8+ T cells polarizing them to the exhaustion state. Many reports suggest that PGE2 changes polarization of T helper cells to different subtypes, alters T cell differentiation and plasticity and enhances induction and differentiation of FoxP3 regulatory T cells [68]. Other studies indicate that PGE2 suppresses T cell proliferation, accelerates T cell replicative senescence markers and induces T cell anergy [69], which could be a sign of exhaustion or induction of anergy mediated by PGE2. In concordance with such observations, our results allow suggesting a new mechanism of immunosuppression as a results of tumor-PNS interactions…

“Please make sure the text is readable and of good quality in all figures. Also, enhance the quality of figures.”

All mistakes in the test and during the loading process have been corrected. All figures are available in high quality for publication.

“Supplementary data

Authors should put the ligands next to represent the figure, not on a separate page.”

Supplemental data have been updated and corrected.

“It is better to provide the uncropped original image of gels/blots with the marker (authors may highlight the cropped portion(s) by line/box etc.).”

All available original data have been loaded.

“Original image of Figure A (pERK1/2 and tERK1/2) is missing.”

We are sorry for another loading mistake.

Round 2

Reviewer 1 Report

Thanks for addressing most of the concerns. 

Author Response

We thank Reviewer 1 for accepting our answers and corrections in the manuscript.

Reviewer 2 Report

The authors answered most of my questions. There is only a small point, if the authors can solve it, I will suggest this article can be accepted in principle. 

For question 1 in the 1st revision, the authors have given some answers. The authors should organize and include the answers in the discussion part of the manuscript.

Author Response

We thank Reviewer 2 for suggesting to discuss limitations of our results in Discussion. We have additionally modified this section of the manuscript and explained that utilization of only melanoma cell lines is an obvious weakness of our model system. We also added a discussion about further studies that should improve our understanding of cancerous cell-Schwann cell crosstalk and provide new data on the mechanisms of tumor-neuro-immune axis function in cancer.